# Three-Dimensional Mapping of Clay and Cation Exchange Capacity of Sandy and Infertile Soil Using EM38 and Inversion Software

**DOI:** 10.3390/s19183936

**Published:** 2019-09-12

**Authors:** Tibet Khongnawang, Ehsan Zare, Dongxue Zhao, Pranee Srihabun, John Triantafilis

**Affiliations:** 1School of Biological, Earth and Environmental Sciences, Faculty of Science, UNSW Sydney, Kensington, NSW 2052, Australia; 2Land Development Regional Office 5, Land Development Department, Khon Kaen 40000, Thailand

**Keywords:** Three-dimensional mapping, quasi-3D inversion algorithm, cation exchange capacity, clay content, sandy infertile soil

## Abstract

Most cultivated upland areas of northeast Thailand are characterized by sandy and infertile soils, which are difficult to improve agriculturally. Information about the clay (%) and cation exchange capacity (CEC—cmol(+)/kg) are required. Because it is expensive to analyse these soil properties, electromagnetic (EM) induction instruments are increasingly being used. This is because the measured apparent soil electrical conductivity (EC_a_—mS/m), can often be correlated directly with measured topsoil (0–0.3 m), subsurface (0.3–0.6 m) and subsoil (0.6–0.9 m) clay and CEC. In this study, we explore the potential to use this approach and considering a linear regression (LR) between EM38 acquired EC_a_ in horizontal (EC_ah_) and vertical (EC_av_) modes of operation and the soil properties at each of these depths. We compare this approach with a universal LR relationship developed between calculated true electrical conductivity (σ—mS/m) and laboratory measured clay and CEC at various depths. We estimate σ by inverting EC_ah_ and EC_av_ data, using a quasi-3D inversion algorithm (EM4Soil). The best LR between EC_a_ and soil properties was between EC_ah_ and subsoil clay (R^2^ = 0.43) and subsoil CEC (R^2^ = 0.56). We concluded these LR were unsatisfactory to predict clay or CEC at any of the three depths, however. In comparison, we found that a universal LR could be established between σ with clay (R^2^ = 0.65) and CEC (R^2^ = 0.68). The LR model validation was tested using a leave-one-out-cross-validation. The results indicated that the universal LR between σ and clay at any depth was precise (RMSE = 2.17), unbiased (ME = 0.27) with good concordance (Lin’s = 0.78). Similarly, satisfactory results were obtained by the LR between σ and CEC (Lin’s = 0.80). We conclude that in a field where a direct LR relationship between clay or CEC and EC_a_ cannot be established, can still potentially be mapped by developing a LR between estimates of σ with clay or CEC if they all vary with depth.

## 1. Introduction

Most cultivated upland areas of northeast Thailand are being used for cash crops (e.g., sugarcane) [1]. However, the soil is sandy and infertile, and they are difficult to improve agriculturally without information about clay and cation exchange capacity (CEC—cmol(+)/kg). In terms of clay, knowledge is important because it is an indication of the capacity of soil to hold moisture and potential to store exchangeable cations [2,3]. Knowledge of the CEC is also necessary because it is a measure of nutrient availability and how well soil pH is buffered against acidification [4] as well as an index of the shrink–swell potential of soil [5]. Therefore, information about the spatial distribution of clay and CEC are required. This is particularly the case in Khon Kaen Province, where poor water holding capacity leads to deep drainage and in some cases rising water tables and soil salinization. In addition, clay (Table 1) and CEC (Table 2) data provides a farmer with information from which fertilizer recommendations can be made.

However, the conventional ways of measuring these soil properties are costly and time-consuming owing to the soil sampling and laboratory analysis. Nevertheless, much research has shown that if many soil samples can be collected, clay and CEC can be mapped using classical geostatistical methods [6,7]. Among the first to map topsoil (0–0.3m) clay and CEC in this way where [8], who used punctual kriging of soil sample locations at the field scale. More recently, [9] predicted topsoil (0–0.15 m) and subsurface (0.3–0.5 m) CEC using various types of kriging (i.e., ordinary) across a large area of North Dakota, USA. Similarly, [10] used additive and modified log-ratio transformation of soil particle size fraction (psf) using ordinary kriging. They then compared this to the untransformed psf data using various kriging techniques (i.e., compositional ordinary- and ordinary-kriging) to predict the topsoil (0–0.1 m) clay, across a very large area in south-eastern Australia. However, a major disadvantage of such geostatistical approaches is that many samples (>100) are required, which need to be spatially correlated and variable [11,12] to yield good results.

To add value to the limited soil data, pedotransfer functions can be used to predict one soil property from another [13,14]. However, to account for short scale variation, easier to acquire ancillary data, which are directly related to clay or CEC are increasingly being used. One of the most widespread are electromagnetic (EM) instruments (i.e., EM38 and EM34), because they measure apparent electrical conductivity (EC_a_—mS/m). [15] were among the first to identify a linear regression (LR) between EM34 EC_a_ and average (0–15 m) clay (R^2^ = 0.73). [16] developed a LR between EM38 EC_a_ and average (0–1.5 m) clay (R^2^ = 0.77) to map clay across a cotton field (244 ha). [17] similarly found a good LR (R^2^ = 0.76) and mapped clay across different fields. In their comprehensive review, [18] demonstrated many other LR of variable strength (R^2^ = 0.01–0.94). In terms of CEC, [19] found a LR between topsoil (0–0.2 m) CEC and EC_a_, while [20] found a strong LR (R^2^ = 0.74) between an EM38 and topsoil (0–0.3 m) CEC across various fields. [21] showed how a LR between an EM38 and average (0–0.2 m) CEC (R^2^ = 0.81) could then be used to map CEC, while [12] established a separate LR to map different topsoil (0–0.075, 0.075–0.15 and 0.15–0.3 m) CEC across a field in Missouri, USA. Again, [18] provided another example of LR between EC_a_ and CEC (0.50–0.76 m).

Given the sandy and infertile nature of soil in northeast Thailand, chemical and compost fertiliser application guidelines [22,23] have been developed. For example, if clay (%) is known and is small (<15%), the chemical fertiliser rates for nitrogen (N), phosphorus (P_2_O_5_) and potassium (K_2_O) would be 113, 38 and 113 kg/ha, respectively. Alternatively, a compost fertiliser rate of 25 t/ha is suggested. This is similarly the case for CEC. In this research our interest is seeing if we can assist farmers with applying these guidelines by developing digital soil maps (DSM). The first aim is to see if we can develop a LR relationship between EM38 EC_a_ directly with either topsoil (0–0.3 m), subsurface (0.3–0.6 m) or subsoil (0.6–0.9 m) clay and CEC. We compare this approach with a universal LR we develop between the calculated true electrical conductivity (σ—mS/m) and laboratory measured clay and CEC at various depths, because of recent success in mapping salinity [24] and moisture [25] by inverting EC_a_ data. While a similar approach was used to map CEC in 3-dimensions by [26], they used a Veris-3100 instrument. Herein, we validate the universal LR using a leave-one-out-cross-validation, considering accuracy, bias and Lin’s concordance. 

## 2. Materials and Methods

### 2.1. Study Area

The study site (Lat 16°11′40.79″ N and Lon 102°43′54.46″ E) is located in the Ban Haet district, Khon Kaen (Thailand). It is situated a short distance to the west of Ban Haet village and located approximately 40 km south of Khon Kaen. The area is approximately 6 ha (150 m × 400 m), with the dominant soil type being an acid sandy loam to sandy Alfisols, described by Land Development Department of Thailand (scale of 1:25,000). The current land use is rain-fed sugarcane farming [1]. The topography across the site is flat to relatively flat with a slope of 0–1%.

The climate is tropical savanna [27]. The mean annual precipitation is around 1,100 mm with the average minimum and maximum temperatures of 18.7 and 35.2 °C, respectively [28]. However, the area has three distinct seasons. The dry-season occurs between mid-February to mid-May with the hottest temperatures in April (43.9 °C) with some rainfall (224.4 mm). Conversely, the rainy season is between May to October with average temperatures typified by July (24.4 °C) which also has the most rainfall (1104 mm). The winter season is between mid-October to mid-February, with maximum temperatures in December (24.2 °C) with limited rainfall (76.3 mm) [28].

### 2.2. Data Collection and Interpolation

The instrument used to collect EC_a_ data was an EM38 [30]. The instrument consists of a transmitter and receiver coil located at either end and spaced 1.0 m apart. The depth of exploration depends on coil configuration. In the horizontal mode, the EM38 measures EC_ah_ and within a theoretical depth of 0–0.75 m. In the vertical mode, the EM38 measures EC_av_ and within a theoretical depth of 0–1.5 m.

In terms of collecting EM38 EC_a_ data in these two modes, 17 parallel transects were defined and spaced approximately 10 m apart in essentially an east–west orientation. Figure 1b shows the spatial distribution of these transects, which were of unequal length. The survey was conducted on 17 January 2018. In all, 467 measurement sites were visited to measure the EM38 EC_ah_ and EC_av_. All EC_a_ measurements were georeferenced using a Garmin Etrex Legend G [31] submeter GPS.

### 2.3. Soil Sampling and Laboratory Analysis

To determine if a direct linear relationship LR between EC_a_ or σ could be developed with topsoil (0–0.3 m), subsurface (0.3–0.6 m) and subsoil (0.6–0.9 m) clay or CEC, 46 soil sampling locations were selected. The sampling points were selected according to two criteria, as suggested by [21]. Firstly, locations with small, intermediate and large EC_a_ were selected; and secondly, samples were spaced evenly across the field. The samples were collected on the 25 January 2018. Figure 1c shows the location of the 46 sampling locations.

The soil samples were air-dried, ground and passed through a 2-mm sieve. Laboratory analysis involved determination of soil particle size fractions (e.g., clay—%) based on Hydrometer method [32]. The cation exchange capacity (CEC—cmol(+)/kg) was also determined based on ammonium saturation method. Regarding this, soil samples were saturated by NH4OAc pH 7.0 and rinsed with NH_4_^+^ using NaCl (Na^+^). Distillation apparatus and titrate method was used to determine CEC [33].

### 2.4. Quasi-3D Inversion of EM38 

The EM4Soil (v304) inversion software package [34] was used to invert EM38 EC_ah_ and EC_av_ to calculate true electrical conductivity (σ—mS/m) and develop electromagnetic conductivity images (EMCI). The quasi-3D inversion algorithm was used in this study. In brief, quasi-3D is a 1-dimensional spatial constrained technique and a forward modelling approach. It assumes that below each EC_a_ measurement location, an estimate of the 1-dimensional variation of σ is constrained because the EM4Soil software considers neighboring locations where the EM38 EC_a_ was measured [35]. 

The raw EC_a_ data was first gridded using a nearest neighbor technique onto a grid spacing of 10 × 10 m using the gridding tool available in EM4Soil. The initial model of σ was set equal to 10 mS/m with the maximum number of iterations equal to 10. A homogeneous five-layer initial model was also considered with depths to the top of each layer being 0, 0.3, 0.6, 0.9 and 1.05 m. The same depths would be used by the EM4Soil software to estimate true σ at these depths and for 3D prediction.

To identify the best possible LR between σ and measured clay and/or CEC, there are a number of other parameters which need to be considered, including selection of a forward model (S1, S2), inversion algorithm (cumulative function (CF) and full solution (FS)), and damping factor (λ). With respect to the inversion algorithm there are two variations (S1 and S2) of Occam’s regularization [36]. The S2 algorithm constrains the model response (of σ) to be around a reference model. It produces therefore smoother results than that of S1. 

Theoretically, the CF model is based on the EC_a_ cumulative response and is used to convert depth profile conductivity to σ [37] considering the condition of low induction numbers. The FS model is based on the Maxwell equations [38] and is not limited to the small induction number condition. Therefore, the FS can improve models calculated from EC_a_ data acquired over highly conductive soils (i.e., >100 mS/m). The damping factor (λ) was progressively increased with smaller increments initially and in large increments thereafter. The λ values used in this study were 0.07, 0.3, 0.6 and 0.9 to balance between rough and smooth EMCIs. 

### 2.5. Validation and Comparison with LR

We use a simple linear regression (LR) model which is in the form of: Y = a + bX + ɛ(1)
where Y is vector of the target property (i.e., clay and CEC) and X is a vector of a predictor while ɛ is the model’s residual. 

Figure 2 shows the flow chart of the two different approaches we undertook. First, we looked to see if six independent MLR models that can be developed between the EC_a_ (i.e., EC_ah_ and EC_av_) data and clay and CEC in either the topsoil (0–0.3 m), subsurface (0.3–0.6 m) and subsoil (0.6–0.9 m). Secondly, we look to see if a satisfactory LR model can be developed between true electrical conductivity (σ—mS/m) as inverted from EM38 EC_ah_ and EC_av_, and clay (%) and CEC (cmol(+)/kg) at all depths using a universal LR, which is applicable at any depth.

To determine the robustness of the calibration and the prediction of either clay and CEC, we tested the final DSM using a leave-one-out-cross-validation procedure. This was carried out 46 times and involved the removal of each of the 46 soil sample locations one at a time and from all three increment depths, including topsoil, subsurface and the subsoil.

The accuracy assessment of prediction was examined using root mean square error (RMSE), whereby the closer the RMSE to zero the more accurate the prediction. The prediction bias was estimated by calculating mean error (ME). Again, the closer to zero then the less biased the prediction. 

The Lin’s concordance correlation coefficient ($\rho_{c}$) was also calculated to assess how close the LR model is to the 1:1 relationship overall. This is because Lin’s concordance correlation coefficient [39] measures degree of agreement between two variables. The Lin’s concordance correlation coefficient is determined from a sample as follows:(2)$$\rho_{c}=\frac{2S_{\mathrm{XY}}}{S_{X}^{2}+S_{Y\;}^{2}+{\left(\bar{X}-\bar{Y}\right)}^{2}}$$
where $\bar{X}\;$and $\bar{Y}$ are means for the two variables (which in our case are the measured and predicted clay or CEC in the topsoil, subsurface or subsoil and $S_{X}^{2}$ and $S_{Y}^{2}\;$are the corresponding variances and
(3)$$S_{\mathrm{XY}}=\frac{1}{n}\overset{n}{\underset{i-1}{\sum }}\left(X_{i}-\bar{X}\right)\left(Y_{i}-\bar{Y}\right)$$

### 2.6. Prediction Interval (PI)

To evaluate the uncertainty, the 95% prediction interval (PI) was used to compute the data between the measured and predicted clay and CEC across the study field and in the topsoil, subsurface and subsoil. The PI represents the frequency of possible confidence intervals that contain the true value of the prediction. A broad PI suggests a larger confidence in prediction [40]. 

## 3. Results and Discussion

### 3.1. Preliminary EC_ah_ and EC_av_ Data Analysis

Table 3 shows the summary statistics of the 467 EC_a_ measured sites during the EM38 survey. The mean EC_ah_ (0–0.75 m) was 23.1 mS/m with a minimum of 14 mS/m and maximum of 35 mS/m. The median (23) was close to the mean, with the EC_ah_ slightly positively skewed (0.2) with a coefficient of variation (CV) of 19.4%. In comparison, the EC_av_ (0–1.5 m) had a larger mean (28.8 mS/m) with a minimum of 18 mS/m and maximum of 45 mS/m. The median EC_av_ was again slightly larger (29.0 mS/m) than the mean with the skewness positive again (0.3) and CV slightly smaller (20.5%). 

Similarly, the simple statistics of the EC_a_ data at the 46 calibration locations were relatively close to the surveyed data. The mean EC_ah_ was 22.3 mS/m with a minimum of 15 mS/m and maximum of 33 mS/m. The median was close to the mean (22), with the EC_ah_ slightly positively skewed (0.4) and with a coefficient of variation (CV) of 19.5%. In comparison, the EC_av_ had a larger mean (27.5 mS/m) with a minimum of 19 mS/m and maximum of 42 mS/m. The median EC_av_ was the same value (27.5 mS/m) to mean with the skewness positive again (0.67) and CV slightly smaller (19.5%).

Figure 3a shows the interpolated digital elevation model (DEM). The highest elevation was in the east end of study field (169 m). The elevation gradually decreased toward the south end of the study field where it was lowest (161 m). Figure 3b shows the interpolated contour plot of measured EC_ah_. The study field was characterized by intermediate-small (15–25 mS/m) EC_ah_ in the northern half. Whereas, intermediate-small to intermediate EC_ah_ (25–35 mS/m) defines the southern.

Figure 3c shows the contour plot of measured EC_av_. Again, the study field was characterized by intermediate-small EC_av_ in the northern half which started from the east through the north west corner and intermediate-large EC_av_ (35–45 mS/m) in the west. From Figure 3b,c and Table 3, we surmise that the subsurface and subsoil were likely to be more conductive than the topsoil.

### 3.2. Preliminary Clay and CEC Data Analysis

Table 4 shows the summary statistics of measured clay (%) at the 46 sample locations. The mean topsoil (0–0.3 m) clay was 11.9% with a minimum of 9.4% and maximum of 16.8%. The median was similar (12%) and the skewness and CV were 0.80 and 12.2, respectively. In the subsurface (0.3–0.6 m), the mean, minimum and maximum were all slightly larger (15.6%, 10.6% and 20.6%, respectively). The median was again similar (15.3%) to the mean, with the clay being positively skewed (0.1). In comparison, the subsoil (0.6–0.9 m) mean of 19.1%, minimum (14.1%) and a maximum (23.4%) was larger again. The median was 19% with a positive skewness (0.1).

Table 4 also shows the summary statistics of measured CEC (cmol(+)/kg) at the sample locations. The mean CEC in the topsoil was 3.3 cmol(+)/kg with a minimum of 2.3 cmol(+)/kg and maximum of 5 cmol(+)/kg. The median was slightly smaller (3.2 cmol(+)/kg) than mean and the skewness and CV were 1.3 and 14.8, respectively. The subsurface mean CEC was larger (4.1 cmol(+)/kg) than the topsoil, with the minimum (2.5 cmol(+)/kg) and maximum (6.3 cmol(+)/kg) also larger. Again, the median was 4 cmol(+)/kg with the skewness and CV were being 0.6 and 18.5. As with the subsoil clay, subsoil CEC had a larger mean (4.9 cmol(+)/kg) minimum (3.5 cmol(+)/kg) and maximum (6.7 cmol(+)/kg). The median was smaller (4.7 cmol(+)/kg) than mean with a skewness and CV of 0.7 and 15.1 respectively.

### 3.3. Spatial Distribution of Clay and CEC Data

Figure 4 shows the contour plot of measured clay (%) and CEC (cmol(+)/kg). Figure 4a shows measured clay in the topsoil (0–0.3 m), which was characterised by small clay (< 15%) across the field. Figure 4b shows measured clay in the subsurface (0.3–0.6 m), which was characterised by a slightly larger clay varying between intermediate-small (15–18%) in the middle of the field and intermediate (18–21%) clay along the southern margin and in the west. 

Figure 4c shows measured clay of the subsoil (0.6–0.9 m). It was characterised in the northern half predominantly by intermediate-small clay. In the west and along the southern margin, clay was intermediate to intermediate-large (21–24%). Clearly, clay increases with depth on average.

Figure 4d shows measured CEC (cmol(+)/kg) from the topsoil. Most of the field was characterised by small (<3.8 cmol(+)/kg) to intermediate-small CEC (3.8–4.5 cmol(+)/kg), except the small area in the west where it was intermediate-large (5.3–6 cmol(+)/kg). 

Figure 4e shows measured CEC in the subsurface, which was generally larger and in accord with the areas where clay was also large. Figure 4f shows measured CEC in the subsoil. As with the clay, as shown in Table 5, the CEC increased with depth on average. For the most part, larger clay and CEC were in accord with the increasing EC_ah_ and EC_av_ from north to south as shown in Figure 3b,c, respectively.

### 3.4. Linear Regression of Clay and CEC of Individual Depth Increment and ECa

Figure 5 shows the LR between measured clay and CEC versus EC_a_. Figure 5a shows EC_ah_ and topsoil (0–0.3 m) clay was small (R^2^ = 0.21). Slightly better correlations were achieved between EC_ah_ and subsurface (0.3–0.6 m) and subsoil (0.6–0.9 m) clay (R^2^ = 0.33 and 0.43, respectively). Figure 5b shows EC_av_ and topsoil clay was also small (0.19), with similarly poor correlations achieved between EC_av_ and subsurface (R^2^ = 0.3) and subsoil (R^2^ = 0.42). Figure 5c shows that equivalent results were achieved between EC_ah_ and CEC, however the correlations were larger in the topsoil (0.50), subsurface (R^2^ = 0.47) and subsoil clay (R^2^ = 0.56) as compared to clay. Figure 5d shows again the same trend of correlation, with the linear regression between EC_av_ and measured CEC equivalent to topsoil (R^2^ = 0.51), subsurface (R^2^ = 0.47) and subsoil (R^2^ = 0.56) CEC. We conclude that there was no satisfactory correlation between the measured soil properties and EC_ah_ and EC_av_ and with increasing depths and therefore no valid LR calibrations to predict these soil properties across our study field. We attribute this to the small CV and subtle differences in clay and CEC across the field.

### 3.5. Linear Regression of Clay for All Depth Increment and σ

To determine if a better coefficient of determination (R^2^) could be obtained between clay and σ, we inverted the EM38 EC_ah_ and EC_av_ using EM4Soil. Table 5 shows the R^2^ between estimated σ (mS/m) obtained from EM4Soil (quasi-3D) and measured clay at all depths (i.e., topsoil, subsurface and subsoil). Table 5 shows that with increase in the damping factor (i.e., λ), the correlation between σ and clay decreases, regardless of algorithm (S1 or S2) or forward model (CF or FS). The best coefficient of determination (R^2^ = 0.65) was when S1 algorithm and FS forward model were used with a λ value of 0.07. The σ values calculated using these parameters were selected to establish a linear regression (LR) between σ and clay. This R^2^ was a little smaller to that achieved by [38] who developed a LR (R^2^ = 0.74) between σ and clay along a single transect in a large irrigated area. We attribute this to the fact that we predicted σ using a quasi-3D inversion, compared to the quasi-2D used along the transect. The effect was greater smoothing herein, as a larger number of neighbours were used to estimate σ. 

Figure 6a shows the LR between σ (S1, FS, and λ = 0.07) and measured clay (%). The LR was made using the bivariate fit tool [41] and include fitted lines, associated fitted confidence intervals (confidence limits for the mean value) and individual confidence intervals (confidence limits for individual predicted values). The LR developed was of the form: clay = 6.04 + 0.50σ. 

Table 6 shows the summary statistics, which indicates the LR was statistically significant (<0.0001*). We note that in Figure 6a topsoil (0–0.3 m) σ was smallest (8–22 mS/m), however, subsurface (0.3–0.6 m) σ was intermediate (11–25 mS/m) with subsoil (0.6–0.9 m) σ largest (17–37 mS/m). The increasing σ was a function of EC_av_ being larger than EC_ah_, as shown in Figure 3b,c, respectively. The reason for the increasing σ was due to increasing clay with depth. 

Figure 6b shows the leave-one-out cross-validation applied for clay by removing each of the 46 sampling locations one at a time. Specifically, the topsoil, subsurface and subsoil samples of one of the sample locations. It was apparent that predicted agreed with measured clay. The prediction precision was good as indicated by a small RMSE (1.78%). The bias of prediction was also good since the ME was close to zero (0.25%). The large Lin's concordance (0.78) between measured and predicted clay signifies good agreement, with the coefficient of determination strong (R^2^ = 0.64).

### 3.6. Linear Regression of CEC for All Depth Increments and σ

Using the same approach, we used for clay, we wanted to determine if a satisfactory LR could be established between CEC and σ estimated from the inversion of EC_ah_ and EC_av_ EM38 data and using EM4Soil. Table 5 also shows the coefficient of determination (R^2^) between estimated σ (mS/m) and CEC at all depths (i.e., topsoil, subsurface and subsoil). As was the case for clay, when λ increased the coefficient between σ and CEC decreased, regardless of algorithm (S1 or S2) or forward model (CF or FS). However, this was the not the case for S1, where the best R^2^ (0.68) overall was achieved when the FS forward model was used with a λ value of 0.9. While this coefficient was equivalent to that achieved for clay, [39] managed to develop a LR (R^2^ = 0.89) between σ and CEC in a small field in Spain. We attribute their success to the fact the EC_a_ data, while collected on 12 m transect spacings, was collected continuously (Veris-3100) and gridded onto a 5 × 5 m grid. In addition, the range in CEC (i.e. range) was much larger in Spain (1–23 cmol(+)/kg) than in this field (2.3–6.7 cmol(+)/kg).

Figure 6c shows the LR between σ and measured CEC (cmol(+)/kg) using these parameters, including the fitted lines, confidence curves and individual confidence curves. The LR was CEC = 1.46 + 0.13σ, with the summary statistics (Table 6) indicating it was significant (<0.0001*). We note that topsoil (0–0.3 m) σ was small (1.80–4.25 mS/m), with subsurface (0.3–0.6 m) intermediate σ (2.2–5.0 mS/m) and subsoil (0.6–0.9 m) σ largest (3.8–7.5 mS/m). Again, increasing σ was a function of the EC_av_ (Figure 3c) being larger than EC_ah_ (Figure 3b). As would therefore be expected, the estimates of subsoil σ were larger than topsoil σ.

Figure 6d shows the leave-one-out cross-validation applied by removing each of the 46 sampling locations one at a time. Again, the topsoil, subsurface and subsoil samples of one of the sample locations was removed and this process was repeated once for each sample location. Predicted CEC was in good concordance (Lin’s = 0.80) with measured CEC, with prediction precision (RMSE = 0.53 cmol(+)/kg) and bias (ME = 0.07 cmol(+)/kg), being small and close to zero, respectively. The coefficient of determination was also strong (R^2^ = 0.66).

### 3.7. Digital Soil Maps of Clay and CEC

Figure 7 shows predicted clay and CEC across the study area as 3D models. Figure 7a was generated after applying the LR (clay = 6.04 + 0.50σ) developed between σ and clay shown in Figure 6a. The predictions were then made using the quasi-3D model generated estimates of σ estimated from passing the gridded EC_a_ data collected from the EM38 EC_a_ across the rest of the area and through EM4Soil and the quasi-3D algorithm. Figure 7b was generated after applying the LR (CEC = 1.46 + 0.13σ) developed between σ and CEC shown in Figure 6b, with predictions applied to the quasi-3D modelled EM38 EC_a_ data as described above for clay.

Figure 8 shows the Digital Soil Mapping (DSM) of predicted clay and CEC at the three sampled depths generated from the LR developed between σ (mS/m) and clay and or CEC. Figure 8a shows predicted topsoil (0–0.3 m) clay. The DSM was like the contour plot generated from the 46 sampling points (Figure 4a) alone. Figure 8b shows the DSM for the subsurface (0.3–0.6 m). The northern half was characterised by small (<15%) clay, while the southern half by intermediate-small (15–18%) clay. These predictions appear less consistent with measured subsurface clay (Figure 4b). Figure 8c shows predicted clay for the subsoil (0.6–0.9 m).

Figure 8d shows the DSM of predicted topsoil (0–0.3 m) CEC. The predicted CEC was consistent with measured CEC, which was small (<3.8 cmol(+)/kg) across most of the field. However, the same could not be said about the small parcel of land along the western margin. Here, there was a band of measured CEC, which was intermediate-small (3.8–4.5 cmol(+)/kg) and intermediate (4.5–5.3 cmol(+)/kg). Predicted CEC was small, however, as was predicted across the entire field. 

There were similar inconsistencies in the subsurface (0.3–0.6 m) predicted CEC as shown in Figure 8e. Here the field was divided into the northern half, which was characterised by predicted CEC which was small, whereas in the southern half it was intermediate-small, with small pockets of intermediate CEC. This was not the case for measured CEC (Figure 4e). These results were consistent with the difference between measured and predicted CEC. Similar results were evident for subsoil (0.6–0.9 m) CEC (Figure 8f) and measured CEC (Figure 4f).

### 3.8. Mapping the Prediction Interval (PI) of Predicted Clay and CEC

To better understand the uncertainty in the predicted DSM of clay and CEC, we mapped the prediction interval (PI). Figure 9a shows the PI for predicted topsoil (0–0.3 m) clay. It indicates that in the southern half, the PI was small (1.0%), whereas the northern half was intermediate-small (1.0–1.5%). Figure 9b shows PI calculated for the subsurface (0.3–0.6 m) clay. Here, PI was generally small across the whole field. Of most interest was the wider PI (>1.5%) in the subsoil (0.6–0.9 m) clay shown in Figure 9c. This was particularly the case for the southern half of the field and in the west. 

Figure 9d shows the PI for predicted topsoil (0–0.3 m) CEC. It showed equivalent patterns to clay. However, there were more classes of PI, with more uncertainty evident in the southern half, where PI was small and intermediate-small (0.2–0.3 cmol(+)/kg). Figure 9e shows the PI calculated for the subsurface (0.3–0.6 m) CEC. Here the PI was generally small (<0.2 cmol(+)/kg) in the southern half. Of most interest was the much wider PI (>0.5 cmol(+)/kg) values for subsoil (0.6–0.9 m) CEC shown in Figure 9f. This was particularly the case for the southern half and in the western margin.

We attribute these PI differences to various factors. Generally, the larger PI were associated with predicted clay and CEC in the subsoil, with the next largest PI associated with the topsoil. With respect to the topsoil, we attribute the larger PI for being a function of EC_ah_ and EC_av_ having a much larger theoretical depth of measurement of 0–0.75 and 0–1.5 m, respectively, compared with estimating σ to a depth of 0–0.3 m. Our ability to resolve topsoil σ, and hence predict clay or CEC, was therefore poor. To improve the estimation of σ and perhaps reduce the PI, we could collect EC_a_ at various heights. This was the approach of [42], who showed that in combination with EM31 EC_ah_ and EC_av_, EM38 EC_ah_ and EC_av_ at a height of 0.6 m was optimal to make a LR between σ and CEC to predict CEC at 0.3 m increments and to a depth of 2.0 m along a single transect. 

Herein, short scale variation was also problematic, owing to the fact that we collected EM38 EC_a_ data on an approximate 10 × 10 m grid. To better account for some of the short scale variation, and reduce the PI, the approach of [26] would be appropriate. This is because they collected EC_a_ from a DUALEM-21 instrument which was mobilized and coupled with a GPS and data logging capabilities. This will allow more detailed and closely spaced EC_a_ data to be collected. We believe a similar approach here, instead of the grid (10 × 10 m) we used, will reduce smoothing and improve prediction accuracy.

### 3.9. Soil Improvement Guidelines Based on Clay and CEC Maps

In spite of the uncertainty, the topsoil (0–0.3 m) and subsurface (0.3–0.6 m) clay DSM (Figure 8a and 8b, respectively) are valuable in terms of practical soil improvement. In the areas where clay was small (<15%), that is in the topsoil and the subsurface in northern half of the area, chemical and compost fertiliser application rates can be suggested according to the guidelines given by [22] and [23], respectively. Specifically, as shown in Table 1, for the areas with small (<15%) clay, chemical fertiliser rate for nitrogen (N), phosphorus (P_2_O_5_) and potassium (K_2_O) can be recommended at 113 (kg/ha), 38 (kg/ha) and 113 (kg/ha), respectively. Alternatively, compost fertiliser at the rate of 25 t/Ha can be suggested. In the southern half of the study area, the rate could be reduced, owing to the slightly larger subsurface clay (15–18%) and specifically for chemical fertiliser for N (113 kg/ha), P_2_O_5_ (38 kg/ha) and K_2_O (75 kg/ha), while a compost fertiliser (19 t/ha) could also be considered. 

With respect to the topsoil CEC map (Figure 8d) the small (<3.8 cmol(+)/kg) CEC are equally informative. Overall, the small (<10 cmol(+)/kg) CEC indicates very poor soil fertility and practically no shrink–swell potential [19]. In situations like this, the subsoil is sometimes more resilient than the topsoil and consideration can be given to invert the profile using a moldboard plough. However, here this was not the case, because even though the subsoil (0.6–0.9 m) CEC was larger (i.e., intermediate-small; <4.5 cmol(+)/kg) it was still below this threshold. 

On the other hand, the topsoil map does provide information which could be useful in assisting with the rate of lime application for the purpose of fertilisation. Specifically, and given this part of Northeast Thailand is a sugarcane growing area, the lime application guidelines indicated in Table 2 could be used. This is despite the fact that the guidelines were developed for application in similarly sandy soil and in tropical climates of North Queensland [29]. Given these guidelines and based on the small (<3.8 cmol(+)/kg) topsoil CEC, a rate of 1.25 t/h of lime should be applied across the entire area.

## 4. Conclusions and Discussion

We were unable to develop any satisfactory linear regression (LR) between EC_ah_ and EC_av_ with measured topsoil (0–0.3 m), subsurface (0.3–0.6 m) and subsoil (0.6–0.9 m) clay (%) or CEC (cmol(+)/kg). We attribute this to the small variation in EC_a_ as well as clay and CEC across the study field and at these three depths. However, the estimates of true electrical conductivity (σ—mS/m) generated by inverting EC_ah_ and EC_av_ and using a quasi-3D algorithm (EM4Soil), enabled the development of a universal LR calibration for both clay and CEC and which included the capability to predict both soil properties in the topsoil, subsurface and subsoil. For clay we found the S1 inversion algorithm with full-solution (FS) and using a damping factor (λ) = 0.07 was optimal (R^2^ = 0.65) with the LR expressed as follows: clay (%) = 6.04 + 0.50σ. For CEC the S1 inversion algorithm, full-solution (FS) and a damping factor (λ) = 0.9 was optimal (R^2^ = 0.68) and could be estimated as follows: CEC (cmol(+)/kg) = 1.46 + 0.13σ. 

We were able to predict and subsequently map the spatial distribution of clay and CEC in the topsoil, subsurface, and subsoil. Subsequently, the uncertainty of these maps was assessed using the prediction interval (PI). We attribute the larger PI in the topsoil to be a function of EC_ah_ and EC_av_ having a theoretical depth of measurement of 0–0.75 and 0–1.5 m, respectively. Given that we were estimating σ to a depth of a topsoil, our ability to do this was not completely satisfactory. This was similarly the case for the larger PI associated with the subsoil depth. 

The solution to these problems would be to collect additional EC_a_ data to better estimate σ. Using our existing EM38, we could collect additional EC_a_ at various heights or by collecting additional data with an EM31. This was the approach carried out by [43], who showed that in combination with EC_ah_ or EC_av_ of EM31, and EC_ah_ or EC_av_ of EM38 at a height of 0.6 m was optimal to make a LR with CEC at 0.3 m increments and to a depth of 2.0 m along a single transect. Alternatively, EC_a_ data could be collected using a multiple-coil EM instrument such as a DUALEM-421 as shown by [44,45,46].

## Figures and Tables

**Figure 1 sensors-19-03936-f001:**
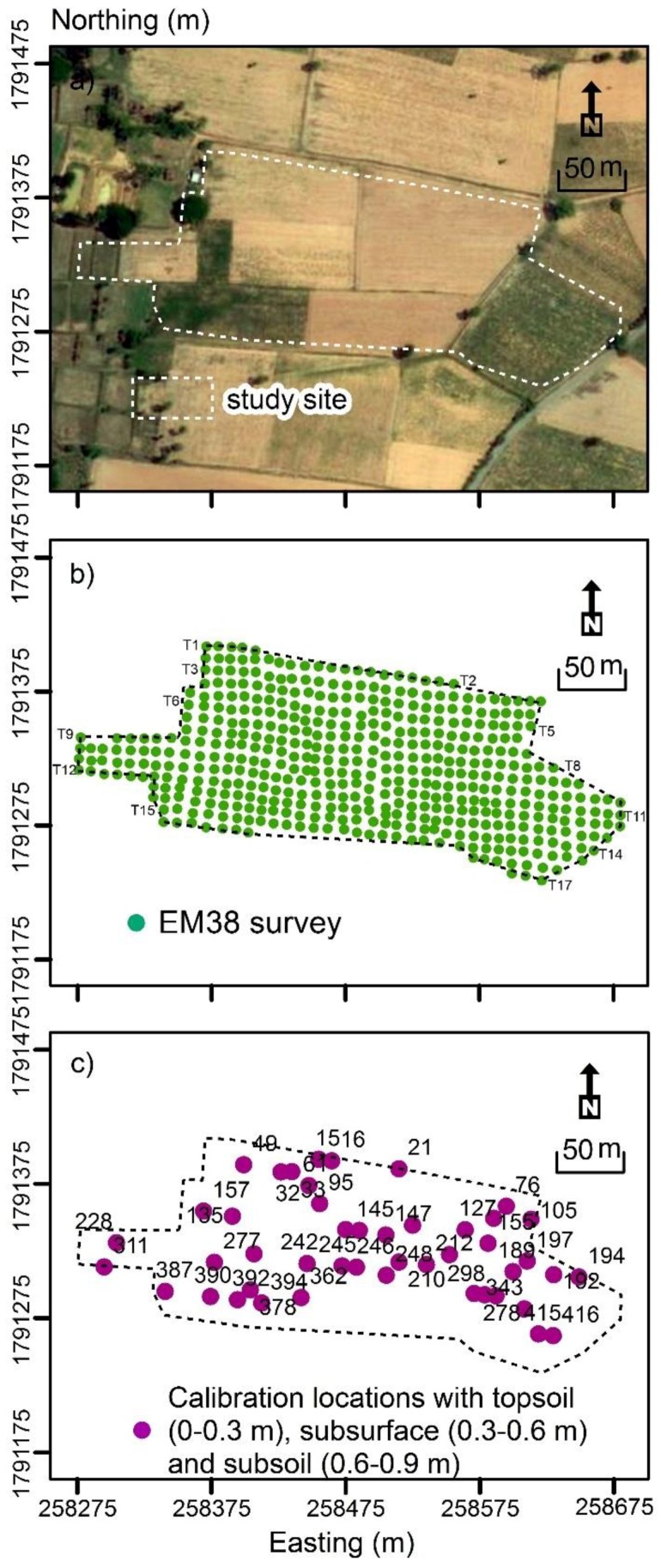
(**a**) Air-photo of study site; (**b**) EM38 survey transects (i.e., 17) and, (**c**) calibration locations (46) where topsoil (0–0.3 m), subsurface (0.3–0.6 m) and subsoil (0.6–0.9 m) samples.

**Figure 2 sensors-19-03936-f002:**
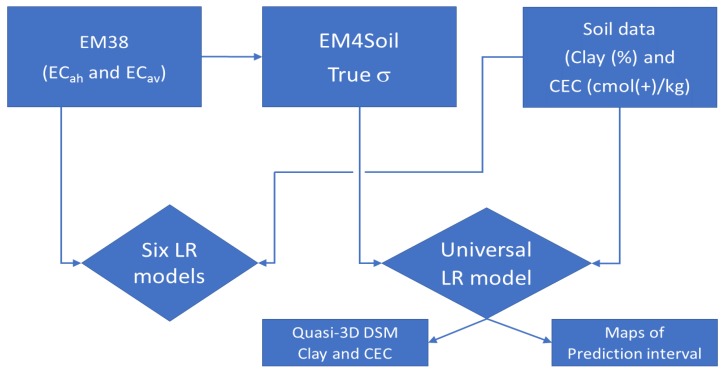
Flow chart of two different approaches to establish a linear regression (LR), between (i) EM38 EC_a_ (mS/m) in horizontal (EC_ah_) or vertical (EC_av_) and three different depths of clay (%) or CEC (cmol(+)/kg) data (i.e., topsoil (0–0.3 m), subsurface (0.3–0.6 m) and subsoil (0.6–0.9m), and (ii) true electrical conductivity (σ—mS/m) inverted from EM38 EC_ah_ and EC_av_ with clay (%) or CEC (cmol(+)/kg) using universal LR at any depth.

**Figure 3 sensors-19-03936-f003:**
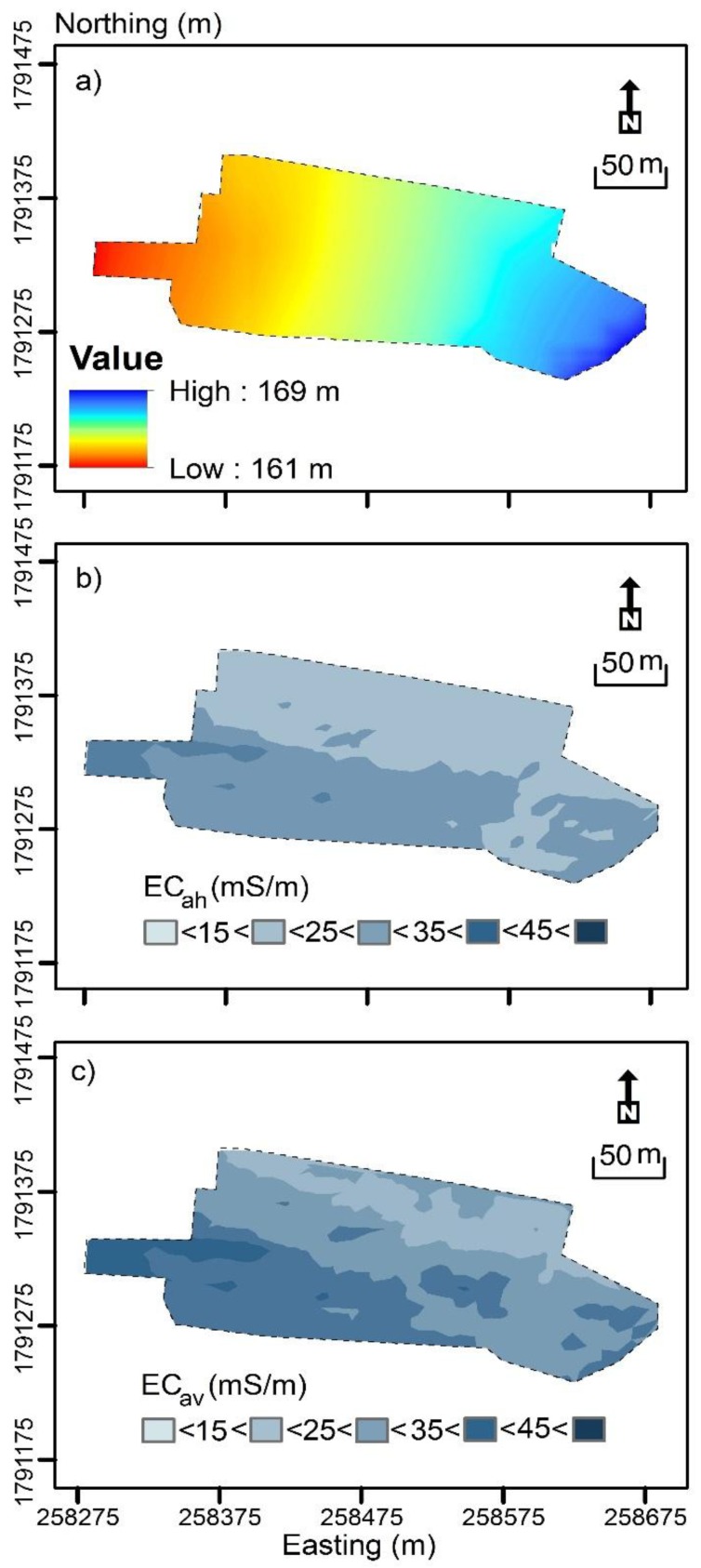
Contour plot of (**a**) elevation (m), and apparent electrical conductivity (EC_a_ – mS/m) of EM38 measured in (**b**) horizontal (EC_ah_; 0–0.75 m), and (**c**) vertical (EC_av_; 0–1.5 m) modes.

**Figure 4 sensors-19-03936-f004:**
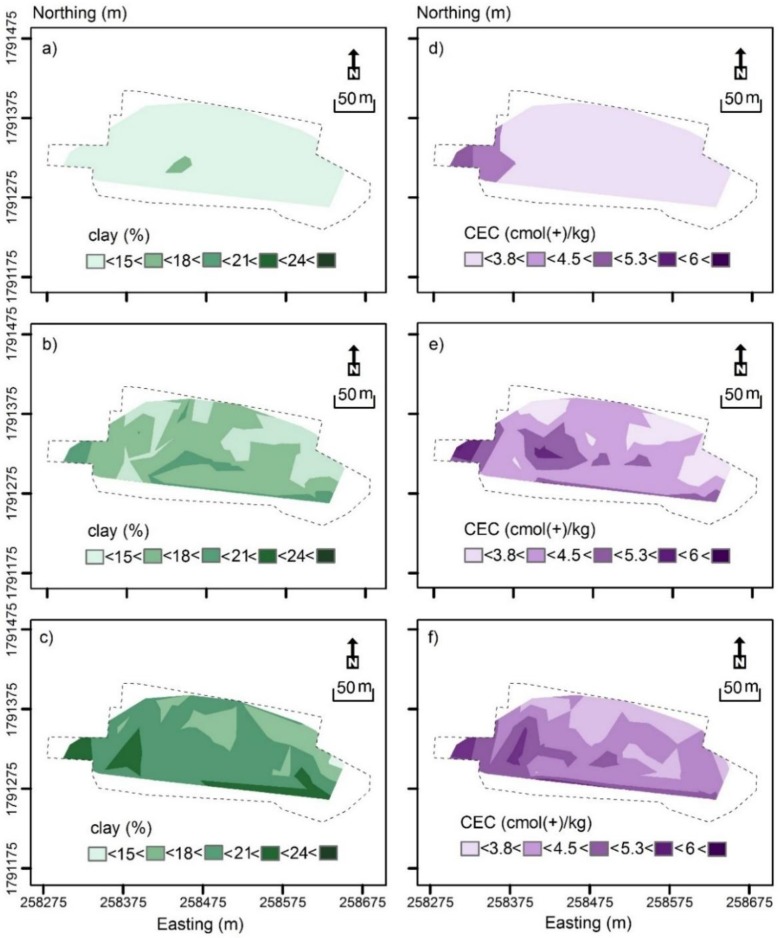
Contour plots of measured (**a**) topsoil (0–0.3 m), (**b**) subsurface (0.3–0.6 m) and (**c**) subsoil (0.6–0.9 m) clay (%) and (**d**) topsoil, (**e**) subsurface and (**f**) subsoil cation exchange capacity (CEC—cmol(+)/kg).

**Figure 5 sensors-19-03936-f005:**
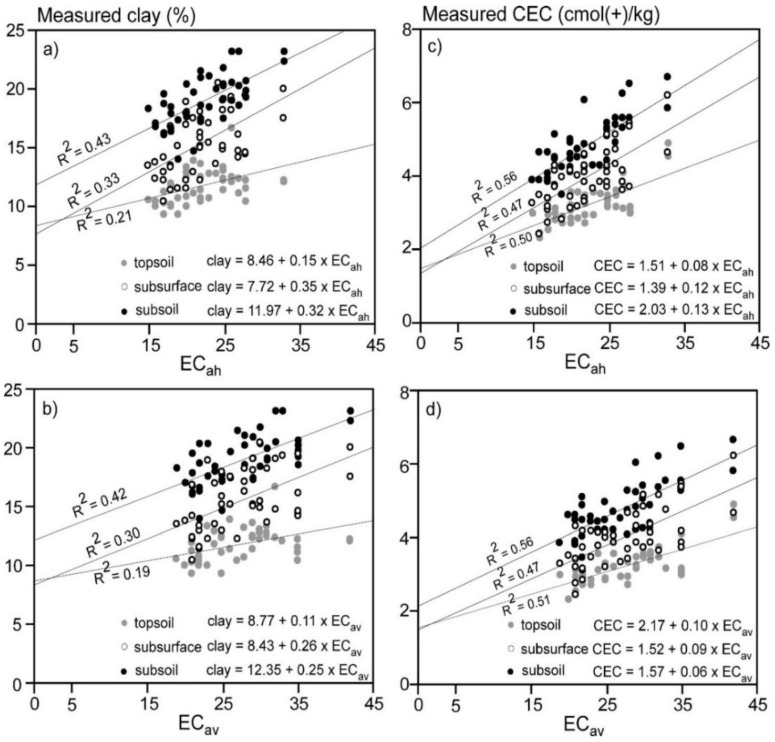
Plots of linear regression (LR) between apparent electrical conductivity (EC_a_—mS/m) measured in (**a**) horizontal (EC_ah_; 0–0.75 m) and (**b**) vertical (EC_av_; 0–1.5 m) modes of EM38 and measured clay (%) in the topsoil (0–0.3 m), subsurface (0.3–0.6 m) and subsoil (0.6–0.9) and plots of LR between (**c**) EC_ah_ and (**d**) EC_av_ and measured cation exchange capacity (CEC—cmol(+)/kg) in the topsoil, subsurface and subsoil.

**Figure 6 sensors-19-03936-f006:**
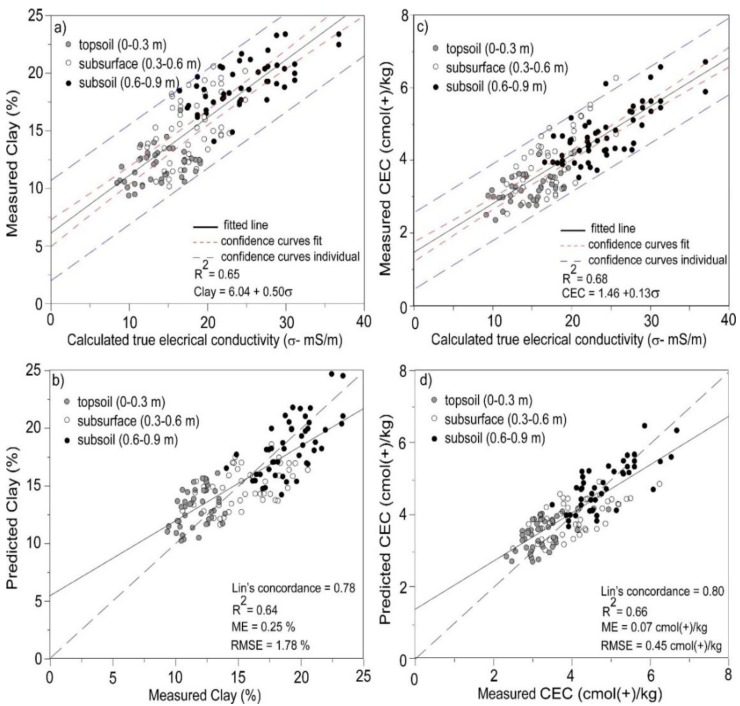
Plots of measured (**a**) clay (%) and (**c**) cation exchange capacity (CEC—cmol(+)/kg) versus true conductivity (σ—mS/m) using λ of 0.07 and 0.9 and S1 inversion algorithm with full solution, respectively; and, plots of measured versus predicted (**b**) clay and (**d**) CEC using leave-one-out cross-validation, for topsoil (0–0.3 m), subsurface (0.3–0.6 m) and subsoil (0.6–0.9 m).

**Figure 7 sensors-19-03936-f007:**
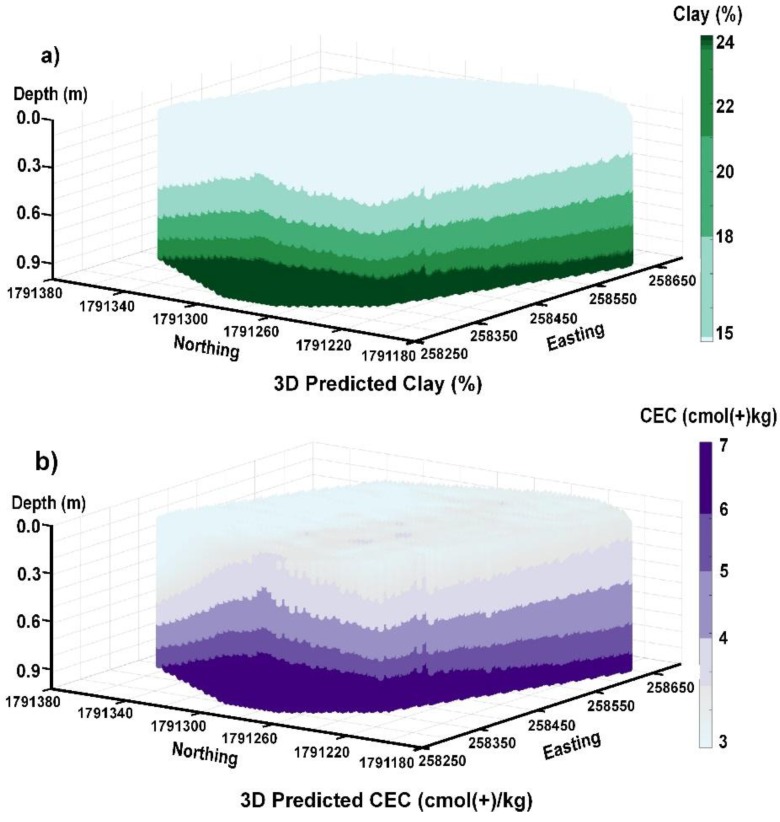
Predicted (**a**) clay (%) and (**b**) CEC (cmol(+)/kg) generated from inversion of EM38 apparent electrical conductivity (EC_a_—mS/m) using EM4Soil and S1 inversion algorithm, full-solution (FS) with damping factor (λ) = 0.07 (clay) and 0.9 (CEC). Note: Calculated true electrical conductivity (σ—mS/m) used to predict clay and CEC from linear regression in Figure 6a,b, respectively.

**Figure 8 sensors-19-03936-f008:**
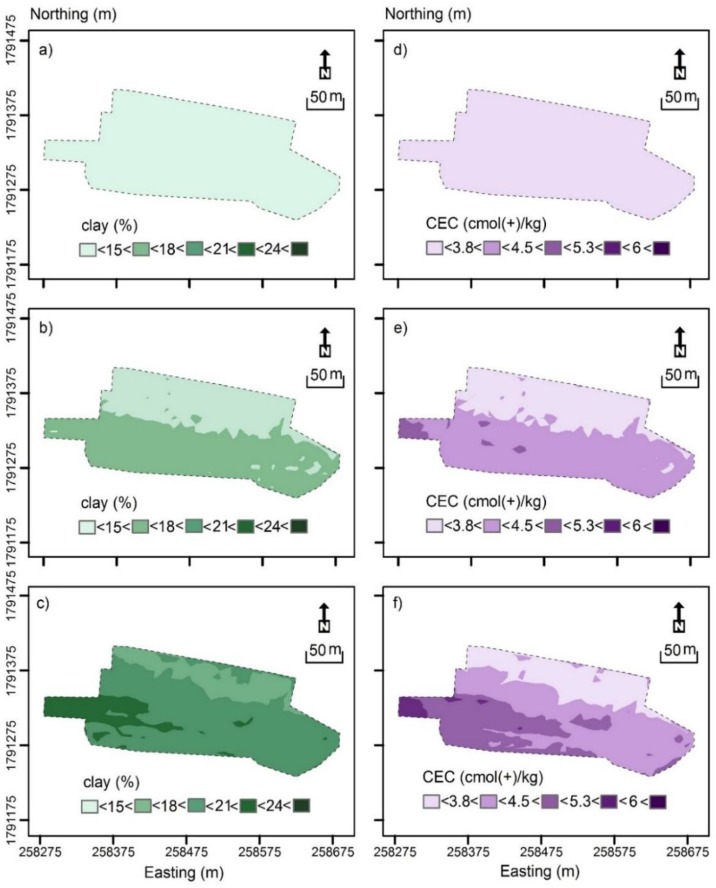
Spatial distribution of predicted clay (%) at depths of (**a**) topsoil (0–0.3 m), (**b**) subsurface (0.3–0.6 m) and (**c**) subsoil (0.6–0.9 m) generated using linear regression (LR) model (Figure 5a), and cation exchange capacity (CEC—cmol(+)/kg) in (**d**) topsoil, (**e**) subsurface and (**f**) subsoil.

**Figure 9 sensors-19-03936-f009:**
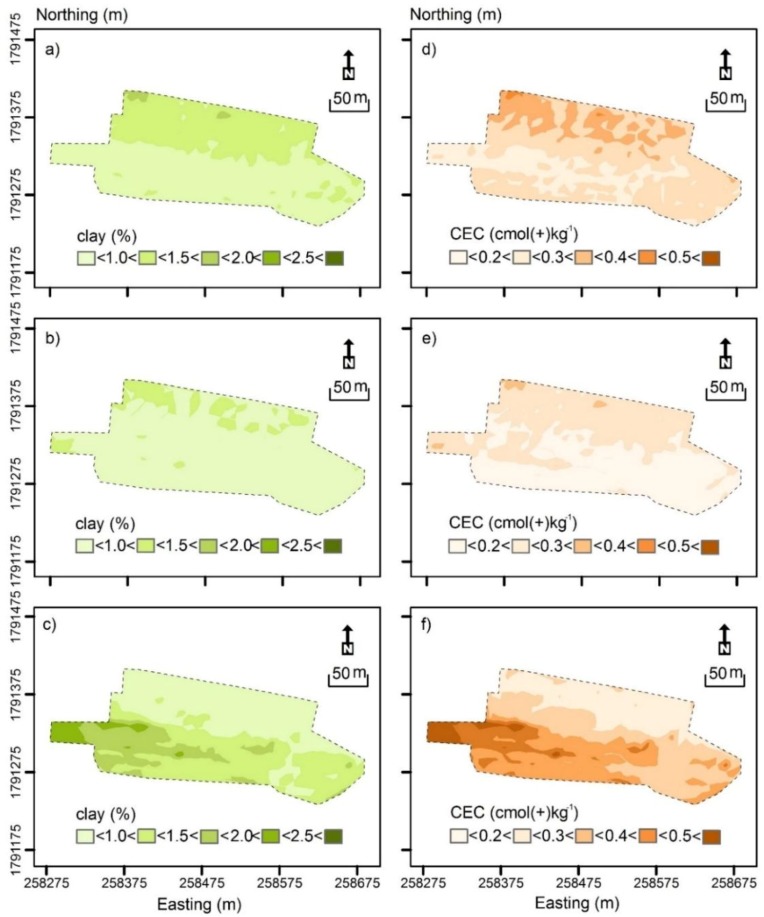
Contour plots of the prediction interval (PI) of the predictions made for clay (%) at different depths including (**a**) topsoil (0–0.3 m), (**b**) subsurface (0.3–0.6 m) and (**c**) subsoil (0.6–0.9 m); and cation exchange capacity (CEC—cmol(+)/kg) in the (**d**) topsoil, (**e**) subsurface and (**f**) subsoil.

**Table 1 sensors-19-03936-t001:** Chemical and compost fertilizer application guidelines based on clay content for sugarcane in Thailand [22,23].

Clay (%)	Chemical Fertilizer Rates (kg/ha)			Compost Fertilizer Rates (t/ha)
	N	P_2_O_5_	K_2_O	
<15	113	38	113	25
15–18	113	38	75	19
18–35	75	19	75	18
>35	72	38	38	18

**Table 2 sensors-19-03936-t002:** Liming application guidelines for sugarcane in Thailand when pH less than 5.0 [29].

CEC (cmol(+)/kg)	Lime Application (t/ha)
<4	1.25
4–8	2.5
8–16	4
>16	5

**Table 3 sensors-19-03936-t003:** Summary statistics of apparent electrical conductivity (EC_a_ mS/m) measured by an EM38 instrument for the entire survey area and at the 46 calibration points.

EC_a_ (mS/m)							
Data Source	n	Min	Mean	Median	Max	Skewness	CV (%)
Survey data							
EC_ah_	467	14	23.1	23	35	0.2	19.4
EC_av_	467	18	28.8	29	45	0.3	20.5
Calibration data							
EC_ah_	46	15	22.3	22	33	0.4	19.5
EC_av_	46	19	27.5	27.5	42	0.67	19.5

**Table 4 sensors-19-03936-t004:** Summary statistics of measured clay (%) and CEC (cmol(+)/kg) at the 46 calibration locations.

Property/Depth	n	Min	Mean	Median	Max	Skewness	CV (%)
clay (%)							
topsoil (0–0.3 m)	46	9.4	11.9	12	16.8	0.8	12.2
subsurface (0.3–0.6 m)	46	10.6	15.6	15.3	20.6	0.1	17
subsoil (0.6–0.9 m)	46	14.1	19.1	19	23.4	0.1	11.2
CEC (cmol(+)/kg)							
topsoil (0–0.3 m)	46	2.3	3.3	3.2	5	1.3	14.8
subsurface (0.3–0.6 m)	46	2.5	4.1	4	6.3	0.6	18.5
subsoil (0.6–0.9 m)	46	3.5	4.9	4.7	6.7	0.7	15.1

**Table 5 sensors-19-03936-t005:** Coefficient of determination (R^2^) achieved between the measured clay and CEC with the estimated true electrical conductivity (σ) generated by inverting EM38 apparent electrical conductivity (EC_a_—mS/m) using the EM4Soil quasi-3D model, cumulative function (CF) or full solution (FS), algorithms S1 or S2, and various damping factors (λ).

**Clay**	**λ**	**S1, CF**	**S1, FS**	**S2, CF**	**S2, FS**
	0.07	0.631	0.648	0.596	0.616
	0.3	0.625	0.643	0.537	0.559
	0.6	0.622	0.643	0.465	0.487
	0.9	0.619	0.637	0.409	0.430
**CEC**					
	0.07	0.666	0.674	0.656	0.666
	0.3	0.666	0.674	0.641	0.655
	0.6	0.667	0.672	0.599	0.617
	0.9	0.668	0.676	0.559	0.577

**Table 6 sensors-19-03936-t006:** Summary statistics of the linear regression (LR) model established between the calculated true electrical conductivity (σ) and measured clay (%) and CEC (cmol(+)/kg). The σ was estimated using the full solution (FS), S1 algorithm and a damping factor (λ) of 0.07 and 0.9 respectively.

**Clay**	**Parameter**	**Estimate**	**SE**	**t-ratio**	**Prob > |t|**	**R^2^**
	Intercept	6.04	0.63	9.62	<0.0001*	0.65
	0.07, S1, FS	0.50	0.03	15.84	<0.0001*	
**CEC**						
	Intercept	1.46	0.16	9.09	<0.0001*	0.68
	0.9, S1, FS	0.13	0.01	16.85	<0.0001*

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
