# Peer review of "Three-Dimensional Mapping of Clay and Cation Exchange Capacity of Sandy and Infertile Soil Using EM38 and Inversion Software"

_sensors, 2019, doi:10.3390/s19183936_

Round 1
Reviewer 1 Report
The paper presents a very interesting subject and scientific approach.
Still I would recommend three little improvements:
- english review - grammar and typos;
- introduction of a methodological diagram - considering that the used methods though interesting but it is hard to follow the different steps.
- conclusion chapter should be strengthened.
Author Response
Dear Editor and Reviewers,
I, on behalf of all authors of this manuscript, express our great appreciation to your valuable work to help us review this paper and provide us with some useful comments and suggestions.
We have revised the manuscript according to all comments and suggestions, which are listed below point-by-point.
We have endeavoured to make our comments in green text and also indicate in blue text how we have changed the manuscript in the accompanying files and based on the suggestions of the reviewers.
We believe the suggestions and comments have helped us to materially improve the scientific quality and merit of the manuscript.
Many thanks go to your valuable help.
Best wishes,
The Authors.
Reviewer 1.
English review - grammar and typosReply
The corresponding author, who has over 100 SCOPUS recognised papers, has thoroughly checked and re-checked for grammatical correctness. Instances where this has been corrected are indicated in green. He has also thoroughly revised the manuscript and re-written sections. In line 423 - the uncertainty of these maps were was assessed using the prediction interval (PI). We have corrected the flawed typos with several locations of symbol of sigma or and ECah ECav to be ECah and ECavintroduction of a methodological diagram - considering that the used methods though interesting but it is hard to follow the different steps.
Reply We agree. We added a working flowchart as shown in the new Figure 3. This flowchart allowed us to better appreciate the methodology. We have added the following text and included the following text in figure 3 caption to explain the Flow chart:
Figure 3 shows the flow chart of the two different approaches we undertook. First, we looked to see if three independent LR models can be developed between the ECa (i.e. ECah and ECav) data and clay and CEC in either the topsoil (0-0.3 m), subsurface (0.3-0.6 m) and subsoil (0.6-0.9 m). Secondly, we look to see if a satisfactory LR model can be developed between true electrical conductivity (s – mS/m) as inverted from EM38 ECah and ECav, and clay (%) and CEC (cmol(+)/kg) at all depths using a universal LR, which is applicable at any depth.
Figure 3. Flow chart of two different approaches to establish a linear regression (LR), between i) EM38 ECa (mS/m) in horizontal (ECah) or vertical (ECav) and three different depths of clay (%) and CEC (cmol(+)/kg) data (i.e. topsoil (0-0.3 m), subsurface (0.3-0.6 m) and subsoil (0.6-0.9m), and ii) true electrical conductivity (s – mS/m) inverted from EM38 ECah and ECav and clay (%) and CEC (cmol(+)/kg) using universal LR.
Conclusion chapter should be strengthened.
Reply - We have rechecked the revised the conclusions. We believe the conclusions are robust and satisfactorily summarise how our results match our stated aims and where improvements in our methodology might yield superior results and to guide future research for others. It now appears as follows:
We were unable to develop any satisfactory linear regression (LR) between either ECah and ECav with measured clay (%) or CEC (cmol(+)/kg). We attribute this to the small spatial variation in ECa as well as clay or CEC across the study field and at these three depths. However, the estimates of true electrical conductivity (s – mS/m) generated by inverting the ECah and ECav and using a quasi-3-dimensional algorithm (EM4Soil), enabled the development of a universal LR calibration for both clay and CEC and which included data for topsoil (0-0.3 m), subsurface (0.3-0.6 m) and subsoil (0.6-0.9 m). For clay we found the S1 inversion algorithm with full-solution (FS) and using a damping factor (λ) = 0.07 was optimal (R2 = 0.65) and estimated as follows; clay (%) = 6.04 + 0.50s. For CEC the S1 inversion algorithm, full-solution (FS) and a damping factor (λ) = 0.9 was optimal (R2 = 0.68) and could be estimated as follows; CEC (cmol(+)/kg) = 1.46 + 0.13s. We were able to predict and subsequently map the 3-d spatial distribution of clay and CEC in the topsoil, subsurface, and subsoil. Consequently, the uncertainty of these maps was assessed using the prediction interval (PI). We attribute the larger PI in the topsoil to be a function of ECah and ECav having a theoretical depth of measurement of 0-0.75 and 0-1.5 m, respectively. Given that, we were estimating s to a depth of a topsoil, our ability to do this was not completely satisfactory. This was similarly the case for the larger PI associated with the subsoil depth. The solution to these problems would be to collect additional ECa data to better estimate s. Using our existing EM38, we could collect additional ECa at various heights or by collecting additional data with an EM31. This was the approach carried out by [40, 41], who showed that in combination with ECah or ECav of EM31, and ECah or ECav of EM38 at a height of 0.6 m was optimal to make a LR with CEC at 0.3 m increments and to a depth of 2.0 m along a single transect. Alternatively, ECa data could be collected using a multiple-coil EM instrument such as a DUALEM-421 and as shown by [31] (pp. 43-53).

Reviewer 2 Report
What authors do isn't real 3D, it is 2.5D (layered) model.
The article looks like the technical report with traditional data sampling technics and statistical methods. Authors apply conventional statistical methods, and results are adequate for the data used, it can be seen from pictures (I also digitize a couple of data sets from pictures, fited linear model, and ran validation protocol - all is OK). The work is well formed case study with a lot of real data and details, and is interesting from the practical point of view - so I recommend to publish it, all is done and presented well (as a practical case study).
From the other hand,
authors don't propose any noticeable innovation (and don't even mention
that neither in abstract nor in conclusion), so I noted the scientific
value as average.
Author Response
Dear Editor and Reviewers,
I, on behalf of all authors of this manuscript, express our great appreciation to your valuable work to help us review this paper and provide us with some useful comments and suggestions.
We have revised the manuscript according to all comments and suggestions, which are listed below point-by-point.
We have endeavoured to make our comments in green text and also indicate in blue text how we have changed the manuscript in the accompanying files and based on the suggestions of the reviewers.
We believe the suggestions and comments have helped us to materially improve the scientific quality and merit of the manuscript.
Many thanks go to your valuable help.
Best wishes,
The Authors.
Reviewer 2
2.1) What authors do isn’t real 3D, it is 2.5D (layered) model
Reply – We do not agree with this statement. What is novel is that we have demonstrated that the use of estimates of calculated sigma from the quasi-3d inversion of the EM38 ECa data can be used to make a universal LR between sigma and either clay and CEC at the same depths and this relationship can be used to predict these soil properties across the entire study area and at any depth! This is in essence a 3d view of the clay and CEC. We have included Figure 7 which shows how we predicted clay and CEC using the quasi-3d inversion.
We have also added the following Figure caption to Figure 7
Figure 7. Predicted a) clay (%) and b) CEC (cmol(+)/kg) generated from inversion of EM38 apparent electrical conductivity (ECa – mS/m) using EM4Soil and S1 inversion algorithm, full-solution (FS) with damping factor (λ) = 0.07 (clay) and 0.9 (CEC). Note: Calculated true electrical conductivity ( – mS/m) used to predict clay and CEC from linear regression in Fig. 6 a) and b), respectively.
2.2) From the other hand, authors don't propose any noticeable innovation (and don't even mention that neither in abstract nor in conclusion), so I noted the scientific value as average
Reply - The innovation of this research has been given in almost entire parts of this paper. Given that, the three-dimensional map of clay and CEC of infertile soil using EM38 and inversion software.
The referee does not clearly understand what we do or does not know of the literature in the area of using EM inversion to model sigma from ECa. There are few examples of where this has actually been done with clay and CEC at the field scale that we are aware of. Certainly, there are no examples in south-east Asia where this approach may be suitable in landscapes of little variability in clay and CEC across the landscape, but owing to leaching of clay and cations deep into the soil from soil forming processes might be useful.

Reviewer 3 Report
Dear editor,
it was a pleasure to review this interesting paper. I can confirm that there is a high-quality science inside. However, the paper has some issues that must be solved in order to get my acceptation. Firstly, the abstract does not give information about the conclusions and applicability of the paper. Secondly, the introduction is interesting but needs to include some more widespread literature about CEC and clays. I suggested 2 references, but the authors can obviate them if they find better options. This is not relevant to me. Thirdly, the study area must be correctly cited and explained with information about the land uses organic matter, past uses, soil types. It is too worth to assume the soil depths that the authors repeat along with the paper for a 6 ha plot. Why is it representative? Why the samples only one time? Fourthly, there is no discussion. Simply, the authors did not include any references. Finally, the conclusions are too long, they have to reduce them. Please, see more comments in the attached pdf.

Author Response
Dear Editor and Reviewers,
I, on behalf of all authors of this manuscript, express our great appreciation to your valuable work to help us review this paper and provide us with some useful comments and suggestions.
We have revised the manuscript according to all comments and suggestions, which are listed below point-by-point.
We have endeavoured to make our comments in green text and also indicate in blue text how we have changed the manuscript in the accompanying files and based on the suggestions of the reviewers.
We believe the suggestions and comments have helped us to materially improve the scientific quality and merit of the manuscript.
Many thanks go to your valuable help.
Best wishes,
The Authors.
Reviewer 3.
Line 17 this is a big assumption of the soil layer depths... this will be your specific study area right?
Reply – Of course it is a big assumption. This is why we want to see if it works. Of course it will be sight-specific. The EM instruments measure ECa which is a function of differences in clay content, clay type, salt content and also moisture.
Line 19 define that
Reply – We are not sure what we need to define?
Line 24 superscript?
Reply - Yes, we have changed “R2 = 0.56” to R2 = 0.56. (Line 24 with red mark version)
It now reads
“…(R2 = 0.56)…”
Line 31 define before
Reply - as mentioned in the first reply
We believe we define the abbreviations already.
Line 39 reference
Reply - We have added a reference from FAO which describes whic type of crops are grown in the northeast of Thailand [1]. (Line 39 with mark version)
[1] FAO, 2005. Management of Tropical Sandy Soils for Sustainable Agriculture “A holistic approach for sustainable development of problem soils in the tropics” 27th November – 2nd December 2005, Khon Kaen Thailand, FAO, Bangkok. 24 p.
Line 39 in the whole region? this is a big assumption.
Reply – The maps we have seen pertaining to the soil types in northeast Thailand commonly represent large swaths of the area being characterised as sandy and infertile.
Line 45, I suggest checking two interesting references related to this topic: Sulieman et al., 2017 and Khlaledian et al. 2017 both in Catena
Reply – We found only the journal article from Sulieman et all, 2018. It is an interesting paper however it is across a much larger scale and there is no use of any proximal soil sensing data which allows us to cite this paper easily into our narrative.
Line 47 why here the tables?
Reply - We mentioned these tables because in the Table 1 and Table 2, we showed the “Chemical and compost fertilizer application guidelines based on clay content for sugarcane in Thailand” and “Liming application guidelines for sugarcane in Thailand when pH less than 5.0” We later refer to these tables and to make inferences on the application rates required to fertilise the soil based on the predicted clay and CEC digital soil maps. We believe this is as good a place as any to provide the Tables. As it turns out, it enabled the typsetting to be neat and quite tidy.
Line 59 split up this sentence.
Reply - We have split the sentence. (Line 55-59 with mark version)
It now reads;
“[9] used additive and modified log-ratio transformation of soil particle size fraction (psf) using ordinary kriging. They then compared this to the untransformed psf data using various kriging techniques (i.e. compositional ordinary- and ordinary-kriging) to predict the topsoil (0-0.1 m) clay, across a very large area in south-eastern Australia”
Line 63 see the above-mentioned papers.
Reply – We have noted and read the paper, however, while it aims to predict CEC at equivalent depths it is not appropriate to reference it herein as these authors did this across a very large spatial extent and they did not use similar geophysical data derived from proximal soil sensing instruments.
Line 69 too many numbers.
Reply – We are simply quoting from the literature the different R2 values which show that ECa data has proven to be a useful ancillary data set to enable calibration and then prediction of various soil properties we are interested here. However, this was not our experience and the inclusion of these numbers demonstrates to a reader that it may not always necessarily be the case that a linear regression can be established as successfully as in these investigations.
Line 77 these numbers are topsoil for you, but topsoil depth depends on the area.
Reply - These were the measured soil clay and CEC at different depths. We define what we mean and we use these definitions consistently throughout the paper. We are clear, consistent and concise in our narrative accordingly. We have used similar terms to those previously used by others and from our own group here at UNSW and as follows:
Ersahin, S., & Brohi, A. R. (2006). Spatial variation of soil water content in topsoil and subsoil of a Typic Ustifluvent. Agricultural Water Management, 83(1-2), 79-86.
Triantafilis, J., Lesch, S. M., La Lau, K., & Buchanan, S. M. (2009). Field level digital soil mapping of cation exchange capacity using electromagnetic induction and a hierarchical spatial regression model. Soil Research, 47(7), 651-663.
Line 88 goals? hyphothesis?
Reply - The research goals were to predict soil clay and CEC with different ways btw direct and indirect methods. The direct method is the model established by measured soil clay and CEC with EM38 ECah and ECav, compare with indirect method by true electrical conductivity (sigma-mS/m) estimated from EM38 ECah and ECav data correlated with measured soil clay and CEC of sandy and infertile soil profile. The outcome by this research is to use the predicted clay and CEC map as a soil management guidelines for farmers.
Line 92 why is it representative?
Reply - This is because sandy and infertile soils are the major part of soil resources in this region.
Line 94 use an updated international soil classification
Reply - Land development department used soil taxonomy classification system (USDA). Why use another system? What is the advantage or disadvantage?
Line 95 references
Reply – [1] FAO, 2005. Management of Tropical Sandy Soils for Sustainable Agriculture “A holistic approach for sustainable development of problem soils in the tropics” 27th November – 2nd December 2005, Khon Kaen Thailand, FAO, Bangkok. 24 p.
Line 104-108 add sources
Reply - We have added one reference as reference [25]. (Line 103 with mark version)
It now reads
“…average minimum and maximum temperatures are 18.7 and 35.2 oC, respectively [25]”
[25] Thai meteorological department website, 2017. https://www.tmd.go.th/en/.
Line 124 only in one moment? try to justify that
Reply - This is because soil properties like clay and CEC are quite stable and would not change if collected at the same location. So there is no reason to collect replicates?
Line 127, units
Reply – we have added the unit of m unit to the Figure 2a and it now reads, 169 m and 161 m
Line 127 Figure 1. this is only one figure.
Reply – There are subcomponents to Figure 1 (i.e. 1a 1b and 1c)?
Line 167 space?
reply – This space should be true electrical conductivity (s) and we have changed this space to s.
It now reads,
“…developed between s and clay…”
Line 409 too long, try to reduce this part to one paragraph.
We believe we need to make conclusions relative to our aims and objectives and we think we have satisfactorilly achieved this now and as follows in the revised three paragraphs.
It now reads,
“We were unable to develop any satisfactory linear regression (LR) between ECah and ECav with measured topsoil (0-0.3 m), subsurface (0.3-0.6 m) and subsoil (0.6-0.9 m) clay (%) or CEC (cmol(+)/kg). We attribute this to the small variation in ECa as well as clay and CEC across the study field and at these three depths. However, the estimates of true electrical conductivity (s – mS/m) generated by inverting ECah and ECav and using a quasi-3 dimensional algorithm (EM4Soil), enabled the development of a universal LR calibration for both clay and CEC and which included data for topsoil, subsurface and subsoil. For clay we found the S1 inversion algorithm with full-solution (FS) and using a damping factor (λ) = 0.07 was optimal (R2 = 0.65) with the LR expressed as follows; clay (%) = 6.04 + 0.50s. For CEC the S1 inversion algorithm, full-solution (FS) and a damping factor (λ) = 0.9 was optimal (R2 = 0.68) and could be estimated as follows; CEC (cmol(+)/kg) = 1.46 + 0.13s.
We were able to predict and subsequently map the spatial distribution of clay and CEC in the topsoil, subsurface, and subsoil. Consequently, the uncertainty of these maps was assessed using the prediction interval (PI). We attribute the larger PI in the topsoil to be a function of ECah and ECav having a theoretical depth of measurement of 0-0.75 and 0-1.5 m, respectively. Given that, we were estimating s to a depth of a topsoil, our ability to do this was not completely satisfactory. This was similarly the case for the larger PI associated with the subsoil depth.
The solution to these problems would be to collect additional ECa data to better estimate s. Using our existing EM38, we could collect additional ECa at various heights or by collecting additional data with an EM31. This was the approach carried out by [42], who showed that in combination with ECah or ECav of EM31, and ECah or ECav of EM38 at a height of 0.6 m was optimal to make a LR with CEC at 0.3 m increments and to a depth of 2.0 m along a single transect. Alternatively, ECa data could be collected using a multiple-coil EM instrument such as a DUALEM-421 and as shown by [31, 43, 44].”
”

Round 2
Reviewer 3 Report
I consider that authors addressed all the suggestions that I made. However, I do not agree that they cannot add Khaledian et al. 2018 and Suleiman et al. 2018 (Catena) or whatever else about larger scales to introduce the general context. The readers of sensors are not all of them experts in cec and clays. This is necessary to mention this change of scale.
Author Response
Dear Editor and Reviewers,
I am, on behalf of all authors of this manuscript, express our great appreciation to your valuable work to help us review this paper and provide us with some useful comments and suggestions.
We have revised the manuscript according to the 3rd reviewer’s comment by adding two sugessted references and rearranged all references to be correctly as the blue text as below.
“To add value to the limited soil data pedotransfer functions can be used to predict one soil property from another [13, 14]. However, to account for short scale variation, easier to acquire ancillary data, which are directly related to clay or CEC are increasingly being used”
13. Khaledian, Y., Brevik, E.C., Pereira, P., Cerdà, A., Fattah, M.A. and Tazikeh, H. 2017. Modeling soil cation exchange capacity in multiple countries. Catena. 158, 194-200.
14. Sulieman, M., Saeed, I., Hassaballa, A. and Rodrigo-Comino, J. 2017. Modeling cation exchange capacity in multi geochronological-derived alluvium soils: An approach based on soil depth intervals. Catena. 167, 327-339.
Many thanks go to your valuable help.
Best wishes,
The Authors.
